# T³RD: Test-Time Training for Rumor Detection on Social Media

## ABSTRACT

With the increasing number of news uploaded on the internet daily, rumor detection has garnered significant attention in recent years. Existing rumor detection approaches excel on familiar topics since there is enough data (high resource) collected from the same domain for model training. However, they are poor at detecting rumors about emergent events especially those propagated in different languages due to the lack of training data and prior knowledge (low resource). To tackle this challenge, we introduce the Test-Time Training for Rumor Detection (T³RD) to enhance the performance of rumor detection models on low-resource datasets. Specifically, we introduce self-supervised learning (SSL) as an auxiliary task in the test-time training (TTT). It consists of local and global contrastive learning (CL), in which the local CL focuses on acquiring invariant node representations and the global CL focuses on obtaining invariant graph representations. We employ auxiliary SSL tasks for both the training and test-time training phase to give a hint about the underlying traits of test samples, subsequently leveraging this hint to calibrate the trained model for these test samples. To mitigate the risk of distribution distortion in test-time training, we introduce a feature alignment aimed at achieving a balanced synergy between the knowledge derived from the training set and the test samples. The experiments conducted on the two widely used cross-domain datasets show that our proposed model achieves state-of-the-art performance. We also provide abundant ablation studies to verify the effectiveness of our methods.

## CCS CONCEPTS

• **Computing methodologies** → **Neural networks**.

## KEYWORDS

Fake News Detection, Test-time Training, Graph Neural Network

## 1 INTRODUCTION

With the rapid development of the Internet, social media has become a convenient platform for users to access information, express opinions, and communicate. More and more people are keen to participate in discussions about the hot topic on social media and express their opinions. Due to the limited domain expertise and relevant data on the emergent events, many rumors appear. For instance, during the unprecedented COVID-19 pandemic, a false rumor claiming that "5G technology was spreading the virus" rapidly spread through social media, causing enormous damage, including the burning of multiple base stations and posing a severe threat to public safety.

Existing rumor detection methods [2, 8, 24, 25, 33, 38], generally follow the conventional training-test paradigm of deep learning. The rumor detection model is learned on the training set to learn the correlations between the label and the latent features of inputs, and then apply the captured knowledge on the test samples to make classifications. Recent approaches [22] model propagation structure

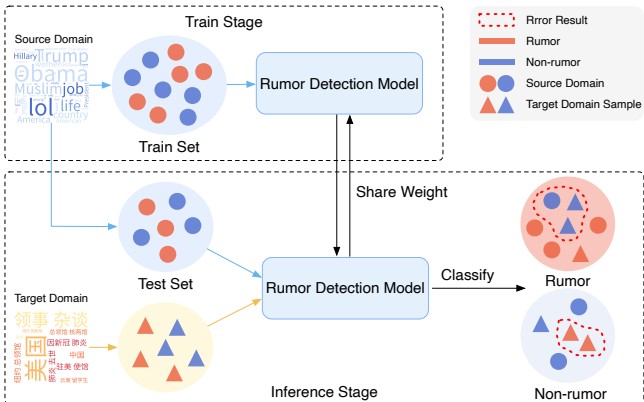

**Figure 1: A toy example to illustrate the gap between source and target domain. The rumor detection models trained on the source domain may not adapt well to the target domain. More error-classifying results are observed when directly inferring the target domain samples on the model.**

as trees to capture structural features. [2] construct graphs for aggregate neighbor features based on reply or retweet relations. [28] jointly model both user and comment propagation networks. However, these approaches are worked underlying a hypothesis that requires the training datasets and test sets to maintain the same distribution. In practical scenarios, real-world news platforms release various claims in different domains every day, and newly emergent and time-critical domain events make it difficult to acquire sufficient labeled data in time. The gap between the source and target domain results in an unsatisfying detection performance. As shown in Fig. 1, the rumor detection model is trained on the source domain, in which the content of training samples is English. When the newly emergent domain is different from the source domain, the model trained on the source domain shows poor performance in the target domain. Directly training the detection model on the source domain and evaluating the newly emergent domain may show an unsatisfactory performance. Some studies[14, 37] try to migrate this challenge by involving some test data in training to learn the knowledge of the target domain.

Test-Time Training (TTT) [27] is the method that can mitigate the distribution shift between source and target domain. The essence of TTT is to design an auxiliary task for both the training and test-time training phase to give a hint about the underlying traits of the target domain samples and then leverage this hint to calibrate the trained model. Considering the unique characteristics exhibited by individual samples from the target domain, TTT has great potential to enhance the generalization capabilities. However, most TTT approaches are designed for the image classification [1, 18, 27] and are not well suited for rumor detection. In rumor detection, limited expertise restricts the acquisition of labeled data

in the target domain. And the data sampled from the target domain is too limited to represent the distribution of the whole test set. In addition, how to alleviate the potential server feature space distortion during test time is another challenge [18] in TTT.

To address these challenges, we present Rumor Detection Test-Time Training($T^3$RD), a novel test-time training framework aimed to enhancing the efficacy of rumor detection in emergency scenarios. Our approach revolves around leveraging graph structures to emulate social media conversations while employing a dedicated hierarchical self-supervised learning framework for test-time training on graphs. In order to mitigate the risk of distribution distortion, we introduce a features alignment that strives to strike a delicate balance between the knowledge derived from the training set and the distinctive signals obtained from a single test sample. Empirical evaluations conducted on various benchmarks demonstrate the superior performance of our proposed method compared to state-of-the-art approaches. Our contributions are summarized as follows:

- We investigate the novel task of test-time training on Rumor detection tasks using social media context, which is capable of simultaneously enjoying the merits of the training knowledge and the characteristics of the test sample.
- We propose a novel model $T^3$RD for test-time training in Rumor detection using social media context, in which various SSL tasks are elaborately designed to boost classification performance and mitigate the potential risk of feature distortion.
- We conduct extensive experiments over two groups of datasets and our proposal consistently achieves superior performance over all the datasets.

## 2 RELATED WORK

**Automatic Rumor Detection.** The initial studies in automatic rumor detection have primarily concentrated on developing a supervised classifier that leverages features extracted from post contents, user profiles, and propagation patterns [3, 17, 34]. Subsequent research has been further advanced by proposing additional features, such as those that capture the characteristics of rumor diffusion and cascades [5, 9, 13]. [39] uses a set of regular expressions to find questing and denying tweets, thereby reducing the engineering effort required. Subsequently, in order to extract features from the continuous flow of social media posts, DNN-based models such as recurrent neural networks [20], convolutional neural networks [35], and attention mechanisms [7] are employed. However, these approaches typically treat the post structure as a simple sequence and overlook the intricate propagation structure present in rumor dissemination.

In the pursuit of extracting valuable cues from both content semantics and propagation structures, several approaches have introduced kernel-learning models [21, 32] to compare propagation trees. To generate representations for each post along a propagation tree, guided by the tree structure, researchers have introduced tree-structured recursive neural networks (RvNN) [23] and transformer-based models [12, 19]. In more recent developments, graph neural networks [2, 15] have been utilized to encode the

conversation thread, resulting in higher-level representations. Nevertheless, these data-driven approaches encounter challenges in detecting rumors in low-resource regimes [11], as they often necessitate substantial training data, which is often scarce in low-resource domains and/or languages. In this paper, we introduce a novel framework that performs model fine-tuning based on each test sample during the test-time training phase, aiming to improve rumor detection in low-resource domains and/or languages.

**Test-Time Training.** The primary objective of test time training is to adapt models based on test samples in the presence of distributional shifts. In [26], the authors introduced a method called test-time training with self-supervision to enhance a model's generalization ability under distribution shift. This is achieved by having the model solve a self-supervised task for test samples. Experimental results in the image domain [1, 6, 26] demonstrate the effectiveness of this framework in reducing the performance gap between training and test sets. There have also been endeavors to integrate test-time training with meta-learning [1], as well as its application in reinforcement learning [10]. Besides these applications, test-time training is also being investigated in the graph domain. In [31], a test-time training framework called GT3 is proposed to bridge the performance gap between the training dataset and the test set by using self-supervised learning at test time to improve node classification performance in the case of data distribution shift. Different from these works of adaption on multi-modal data, we focus on language and domain adaptation to detect rumors from low-resource microblog posts corresponding to breaking events.

## 3 PRELIMINARY

Given a source domain dataset $D^s = \{C_i^s\}, i = [1, \ldots, M], C_i^s = \{c_i, y_i, T(c_i)\}$ is the tuple that consists of the source claim $c_i$, the label $y_i \in \{rumor, non-rumor\}$, and all its responsive posts $T(c_i)$. $T(c_i) = \{c_i, r_1^i, \ldots, r_{|c|-1}^i\}$. $r_j^i$ is the $j-th$ responsive text, and $|c|-1$ is the total number of responsive posts. For the target domain, we consider a much smaller dataset for testing $D^t = \{C_i^t\}, i = [1, \ldots, N](N \ll M)$. $C_i^t$ has a similar composition structure to the source dataset.

Building upon the aforementioned information, we can obtain the propagation structure graph $G_i = < V_i, E_i >$, where $V_i = \{c_i, r_1^i, \ldots, r_{|c|-1}^i\}$ is the node, $c_i$ is the root node, and $E_i = \{e_{he}^i \mid h, e = 0, \ldots, |c|-1\}$ denotes the set of edges from responded posts to the retweeted posts or responsive posts. Considering $r_2^i$ has a response to $r_1^i$, there will be a directed edge $r_1^i \rightarrow r_2^i$, i.e., $e_{12}^i$. Note that $\mathbf{A}_i \in \{0, 1\}^{|c| \times |c|}$ is the adjacency matrix and defined as follows:

$$a_{eh}^i = \begin{cases} 1, & \text{if } e_{he}^i \in E_i \\ 0, & \text{otherwise} \end{cases} \tag{1}$$

$\mathbf{X}_i = [\mathbf{x}_0^{i\top}, \mathbf{x}_1^{i\top}, \ldots, \mathbf{x}_{|c|-1}^{i\top}]^\top$ is the feature matrix that is extracted from the posts in $C_i^*; * \in \{s, t\}$. $\mathbf{x}_0^i$ represents the feature vector of $c_i$ and each other row feature $\mathbf{x}_j^i$ represents the feature vector of $r_j^i$. The target of rumor detection is to train a classification model $f$ that is used to predict whether the claim from the target domain is a rumor.

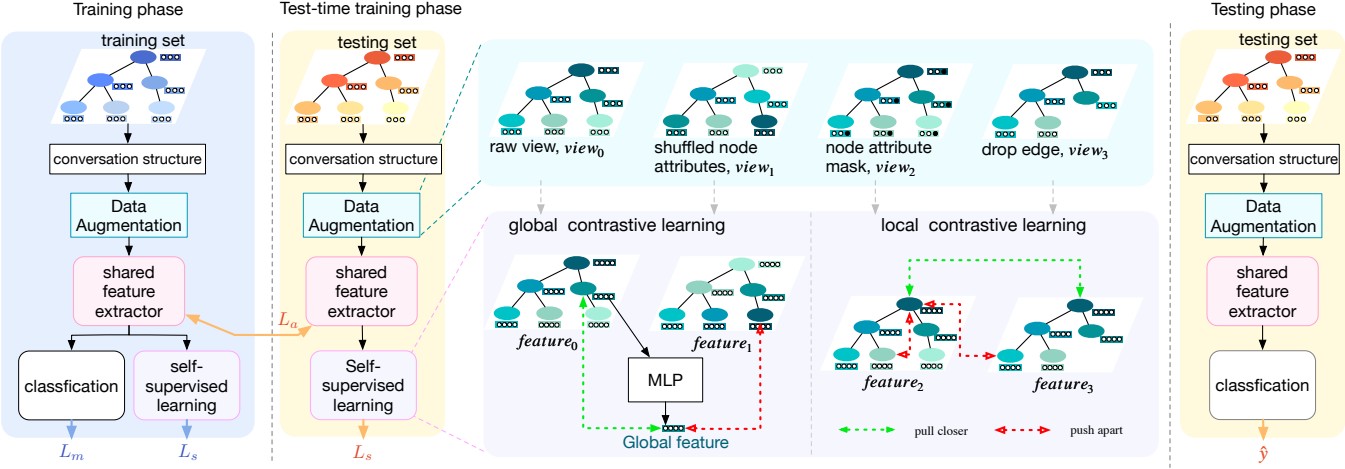

**Figure 2: The overall architecture of T³RD. The shared feature extractor employs GCN to extract input graph features. $L_a$ represents the loss for aligning the feature space. Additionally, the blue $L_m$ and $L_s$ represent the losses obtained during the training phase for the main and auxiliary tasks, while the orange $L_s$ and $L_c$ denote the losses during the test-time training phase. $\hat{y}$ denote the predicted values during the testing phase.**

## 4 METHODOLOGY

In this study on rumor detection in a few-shot scenario within the target domain, we present a rumor detection model trained on ample source domain data and a limited amount of target domain data. However, traditional models often prioritize the utilization of the training set while neglecting the test set. Our objective is to thoroughly exploit the information within the test dataset to maximize detection performance. In this section, we present the proposed architecture of Test-Time Training for Rumor Detection (T³RD) and elaborate on the components.

### 4.1 Architecture

First, the objective of this study is rumor detection, which serves as the main task of T³RD. Here, we define the loss function for the primary task as $L_m(\cdot)$ (in Sec. 4.2). Second, as informed by [18], the auxiliary tasks should effectively capture information social media conversation data. To achieve this, we elaborately design a self-supervised contrastive learning approach that captures node-node and node-graph information (in Sec. 4.3). For simplicity, we denote the objective of the SSL task as $L_s(\cdot)$. Third, as mentioned in [18], if a significant distribution shift exists, test-time training may not enhance performance and could potentially worsen it. To address this, we propose feature alignment (in Sec. 4.4), denoting the objective of this task as $L_a(\cdot)$. The overall architecture of our rumor detection approach is presented in Fig. 2. It consists of training, test-time training, and test phase.

*4.1.1 The Training Phase.* Given the training set $D^*, * \in s, t$, during the training phase, we train all parameters for both the main task and the SSL task. Inspired by [31], to effectively integrate the main task and auxiliary tasks of rumor detection, we enable parameter sharing between the two layers of GCN responsible for feature extraction. We denote the shared model parameters as $\theta_e = (\theta_1, \theta_2)$. The model parameters specifically designated for the main task are

denoted as $\theta_m$, corresponding to the parameters in the prediction layer for rumor detection. Correspondingly, the SSL task has its self-supervised learning parameters, denoted as $\theta_s$, which represent the contrastive layer parameters for the SSL task. In summary, the model parameters are as follows: $\theta_{main} = (\theta_e, \theta_m)$ for the main task, $\theta_{self} = (\theta_e, \theta_s)$ for the auxiliary task, $\theta_{align} = (\theta_e)$ for the feature alignment task, and $\theta_{overall} = (\theta_e, \theta_m, \theta_s)$ for the entire framework.

In the training process, we optimize the entire framework by minimizing a weighted combination of losses $L_m(\cdot)$ and $L_s(\cdot)$ as follows:

$$\min_{\theta_e, \theta_m, \theta_s} \frac{1}{n} \sum_{i=1}^{N} L_m\left(\mathbf{A}_i, \mathbf{X}_i, y_i; \boldsymbol{\theta}_e, \boldsymbol{\theta}_m\right) + \alpha L_s\left(\mathbf{A}_i, \mathbf{X}_i; \boldsymbol{\theta}_e, \boldsymbol{\theta}_s\right) \quad (2)$$

where $\alpha$ is a hyper-parameter balancing the rumor detection task and the self-supervised task in the training process.

*4.1.2 The Test-time Training Phase.* Given the test sample $C^t$, the test-time training process fine-tunes the learned model through the SSL task and feature alignment task. Specifically, the shared model parameters $\theta_e$ and the self-supervised learning parameters $\theta_s$ are fine-tuned by minimizing a weighted combination of losses $L_s$ and $L_a$:

$$\min_{\theta_e, \theta_s} L_s\left(\mathbf{A}_t, \mathbf{X}_t; \boldsymbol{\theta}_e, \boldsymbol{\theta}_s\right) + \gamma L_a\left(\mathbf{A}_t, \mathbf{X}_t; \boldsymbol{\theta}_e, \right) \quad (3)$$

where $\gamma$ is a hyper-parameter balancing the self-supervised learning task and features alignment task in the test-time training process. Suppose that $\theta^*_{overall} = \left(\theta^*_e, \theta^*_m, \theta^*_s\right)$ represents the optimal parameters obtained from minimizing the overall loss of Eq. 2 during the training process. In the test-time training phase, the proposed framework further updates $\theta^*_e$ and $\theta^*_s$ to $\theta^{*'}_e$ and $\theta^{*'}_s$ by minimizing the SSL loss and feature alignment loss of Eq. 3 over the test sample $C^t$.

*4.1.3 The Test Phase.* In the testing phase, we predict the labels of $C^t$ using the classification model $f(\cdot; \theta_e^{*\prime}, \theta_m^*)$:

$$\hat{y}_t = f(\mathbf{A}_t, \mathbf{X}_t; \theta_e^{*\prime}, \theta_m^*) \tag{4}$$

## 4.2 Rumor Detection

For a given event $C_i = \{c_i, y_i, T(c_i)\}$, we obtain its adjacency matrix $A_i$ and feature matrix $X_i$. Inspired by [14], here, we first obtain propagation and dispersion features through Top-Down graph convolutional Networks (TD-GCN) and Bottom-Up graph convolutional Networks (BU-GCN), respectively. Then, we use the supervised contrastive learning to align the representation space of rumor-indicative signals from different domains and languages.

**Top-Down GCN.** We utilize a layer-wise propagation rule to update the node vector at the $l-th$ layer:

$$H^{(l+1)} = \text{ReLU}(\hat{\mathbf{A}}_{\mathbf{i}} \cdot H^{(l)} \cdot W^{(l)}) \tag{5}$$

In the above equation $\hat{\mathbf{A}}_{\mathbf{i}} = \tilde{\mathbf{D}}^{-\frac{1}{2}} \tilde{\mathbf{A}}_{\mathbf{i}} \tilde{\mathbf{D}}^{-\frac{1}{2}}$ is the normalized adjacency matrix, where $\tilde{\mathbf{A}}_{\mathbf{i}} = \mathbf{A}_{\mathbf{i}} + \mathbf{I}_N$ (i.e., adding selfconnection), $\tilde{\mathbf{D}}_{ii} = \Sigma_j \tilde{\mathbf{A}}_{ij}$ that represents the degree of the $i-th$ node; $W^{(l)} \in \mathbb{R}^{d^{(l)} \times d^{(l+1)}}$ is a layer-specific trainable transformation matrix. $H^{(0)}$ is initialized as $X_i$. For a GCN model with 2-layers, we obtain the final node representation $H_{TD}$ w.r.t $H^{(2)}$.

**Bottom-Up GCN.** Similar to the Top-Down GCN, we follow the same procedure as described in Eq. 5 to update the hidden representation of nodes, ultimately obtaining the output node states $H_{BU}$ at the $L$-th graph convolutional layer.

**The feature representation.** We aggregate the output of TD-GCN and BU-GCN separately using mean-pooling to obtain representations for propagation and dispersion, and then concatenate these representations:

$$o_i = \text{concat}(\text{mean-pooling}(H_{TD}), \text{mean-pooling}(H_{BU})) \tag{6}$$

**Supervised Contrastive Training.** For an event $C_i^s$ from the source data, to enhance the discriminative power of rumor representations within source events, we introduce a supervised contrastive learning objective aimed at clustering samples of the same class while separating those from different classes.

$$L_{SCL}^s = -\frac{1}{2N^s} \sum_{i=1}^{N^s} \frac{1}{N_{y_i^s} - 1} \sum_{j=1}^{N^s} \mathbb{1}_{[i \neq j]} \mathbb{1}_{\left[y_i^s = y_j^s\right]}$$
$$\log \frac{\exp\left(\text{sim}\left(o_i^s, o_j^s\right)/\tau\right)}{\sum_{k=1}^{N^s} \mathbb{1}_{[i \neq k]} \exp\left(\text{sim}\left(o_i^s, o_k^s\right)/\tau\right)} \tag{7}$$

where $N_{y_i^s}$ represents the number of source examples with the same label $y_i^s$ in the event $C_i^s$, and $\mathbb{1}$ is the indicator. $sim(\cdot)$ refers to the cosine similarity function and $\tau$ is the temperature parameter.

For an event $C_i^t$ from the target data, to align the features of source domain and target domain samples, we employ contrastive learning, which brings samples from the same category in the source and target domains closer together than samples from different categories.

$$L_{SCL}^t = -\frac{1}{2N^t} \sum_{i=1}^{N^t} \frac{1}{N_{y_i^t}} \sum_{j=1}^{N^s} \mathbb{1}_{\left[y_i^t = y_j^s\right]}$$
$$\log \frac{\exp\left(\text{sim}\left(o_i^t, o_j^s\right)/\tau\right)}{\sum_{k=1}^{N^s} \exp\left(\text{sim}\left(o_i^t, o_k^s\right)/\tau\right)} \tag{8}$$

where $N^t$ is the total number of target examples in the batch and $N_{y_i^t}$ is the number of source examples with the same label $y_i^t$ in the event $C_i^t$.

**The overall loss function.** We input the feature representations $o_i$ into a softmax function to obtain the predicted label $\hat{y}_i$. Then, we learn to minimize the cross-entropy loss between the prediction and the ground-truth label $y_i^*, * \in \{s, t\}$:

$$L_{CE}^* = -\frac{1}{2N^*} \sum_{i=1}^{N^*} \log(p_i) \tag{9}$$

where $N^*$ is the total number of source examples in the batch, $p_i$ is the probability of correct prediction. Finally, we jointly train the main task with the cross-entropy and supervised contrastive objectives:

$$L_m = L_{CE}^* + L_{SCL}^*; * \in \{s, t\} \tag{10}$$

## 4.3 Self-supervised Learning

The key to the success of test-time training lies in an appropriate and informative SSL task [18]. However, designing a suitable SSL task for test-time training of rumor detection models is challenging. Specifically, Existing methods that involve test-time training with graph data are relatively limited, and even when available, they primarily focus on graph structures resembling molecular structures. However, the structure of rumor propagation graphs significantly differs from that of molecular structures. Firstly, the number of nodes in rumor propagation graphs greatly exceeds the number of molecular components, posing a challenge for generating propagation graphs node by node. Secondly, molecular graphs typically adhere to specific connection rules, such as atoms in a molecule being frequently connected through covalent bonds. These patterns, however, do not apply to rumor propagation graphs, rendering graph-based methods unsuitable for direct use in rumor detection [4]. Therefore, it is important to take full advantage of the rumor propagation feature in the design of SSL tasks.

Inspired by the success of contrastive learning in graph domains [31], we present a hierarchical SSL task for T³RD that integrates both local and global perspectives. By doing so, we fully leverage the graph information from both the node-node level and node-graph level. Notably, our proposed SSL task is not built relying on differences among different graphs, so it can be applied to a single graph. Our empirical validation confirms the necessity of both the global (node-graph level) contrastive learning task and the local (node-node level) contrastive learning task in the proposed two-level SSL framework for T³RD. In the following sections, we will provide detailed explanations of the global contrastive learning task and the local contrastive learning task.

*4.3.1 Global Contrastive Learning.* The objective of global contrastive learning is to help the node representation in capturing global information from the entirety of the social media conversation graph. In essence, global contrastive learning operates by maximizing the mutual information between the local node representations and the global graph representation. As shown in Fig. 2, given a rumor propagation structure graph $G_i$, different views can be generated with various types of data augmentation techniques. For global contrastive learning, we employ two views: one is the

raw view $View_0$, where the original graph remains unaltered; the other is the augmented view $View_1$, where node attributes are randomly shuffled across all nodes in the graph. Leveraging these two graph views, we can obtain two corresponding node representations, $feature_0$ and $feature_1$, through a shared feature extractor. Following this, a global graph representation is summarized by employing a multilayer perceptron to process the node representation matrix $feature_0$, which is extracted from the raw view $View_0$.

As per [30], the positive samples in the global contrastive learning are composed of node-graph representation pairs, with both the node representation and graph representation originating from the raw view $View_0$. The negative samples are composed of node-graph representation pairs, wherein the node representations are sourced from $View_1$, while the graph representation is extracted from $View_0$. We employ a discriminator $D$ to calculate the probability score for each pair, where positive pairs should have higher scores, and negative pairs should have lower scores. In our work, we define $D(Z_{si}, s) = Sigmoid(Z_{si} * s)$, where $Z_{si}$ signifies the representation of node $i$ from $View_s$, and $*$ denotes the inner product.

The objective function for the global contrastive learning can be summarized as follows:

$$L_g = -\frac{1}{2N}\left(\sum_{i=1}^{N}(\log D(Z_{0i}, s) + \log(1 - D(Z_{1i}, s)))\right), \quad (11)$$

where $N$ represents the number of nodes in the input graph.

*4.3.2 Local Contrastive Learning.* In global contrastive learning, the primary objective of the model is to capture the global information of the entire graph into the node representations. Put differently, the model aims to acquire an invariant graph representation at a global level and integrate it into the node representations. However, graph data is composed of nodes with various attributes and distinct structural roles. To fully harness the structural information within a graph, it is essential not only to obtain an invariant graph representation at a global level but also to learn invariant node representations from a local level. To achieve this, we introduce local contrastive learning, which is grounded in distinguishing diverse nodes from distinct augmented views of a graph.

Given a rumor propagation structure input graph $G_i$, as shown in Fig. 2, two views of the graph can be generated through data augmentation. As local contrastive learning aims to differentiate whether two nodes from different views represent the same node in the input graph, it is imperative that the employed data augmentation does not introduce significant alterations to the input graph. In order to fulfill this requirement, we employ two graph data augmentation mechanisms - edge dropping and node attribute masking. Following data augmentation, we get two views of an input graph, which then serve as the input for the shared GCN model. The outputs consist of two node representation matrices, denoted as $feature_2$ and $feature_3$, which correspond to two views, respectively. Please note that the fundamental objective of local contrastive learning is to discern whether two nodes from augmented views represent the same node in the input graph. Therefore, $(Z_{2i}, Z_{3i})(i \in \{0, \ldots, |c| - 1\})$ represent positive pairs, where $|c|$ is the number of nodes. On the other hand, the pairs $(Z_{2i}, Z_{3j})$ and $(Z_{2i}, Z_{2j})(i, j \in \{1, \ldots, N\} and i \neq j)$ represent intra-view negative pairs and inter-view negative pairs, respectively. Inspired by

**Table 1: Statistics of the datasets in this paper.**

| Statistics | Source Twitter | Target Weibo-COVID19 | Source Weibo | Target Twitter-COVID19 |
|---|---|---|---|---|
| # of events | 1154 | 399 | 4649 | 400 |
| # of tree nodes | 60409 | 26687 | 1956449 | 406185 |
| # of non-rumors | 579 | 146 | 2336 | 148 |
| # of rumors | 575 | 253 | 2313 | 252 |
| Avg.# of posts/tree | 52 | 67 | 420 | 1015 |
| Language | English | Chinese | Chinese | English |

InfoNCE [29, 40], we define the objective for a positive node pair $(Z_{2i}, Z_{3i})$ as follows:

$$I_c(Z_{2i}, Z_{3i}) = \log \frac{h(Z_{2i}, Z_{3i})}{h(Z_{2i}, Z_{3i}) + \sum_{j\neq i} h(Z_{2i}, Z_{3j}) + \sum_{j\neq i} h(Z_{2i}, Z_{2j})} \quad (12)$$

where $h(Z_{2i}, Z_{3j}) = e^{cos(g(Z_{2i}),g(Z_{3j}))/\tau}$, the function $cos()$ represents the cosine similarity. The symbol $\tau$ represents the temperature parameter, and $g()$ denotes a two-layer perceptron (MLP) used to enhance the expressive capacity of the model further. The MLP refined node representations are denoted as $Z_2'$ and $Z_3'$, respectively.

$$L_l = -\frac{1}{2N}\sum_{i=1}^{N}(I_c(Z_{2i}, Z_{3i}) + I_c(Z_{3i}, Z_{2i})) \quad (13)$$

*4.3.3 The Overall Loss Function.* The SSL task total loss function for T³RD is a weighted combination of global and local contrastive learning losses:

$$L_s = L_g + \beta L_l, \quad (14)$$

where $\beta$ is the parameter that balances global and local contrastive learning. Through minimizing the SSL loss defined in Eq. ??, the model simultaneously captures global graph information and learns invariant and crucial node representations.

## 4.4 Feature Alignment

Directly applying test-time training, the model may overfit to the SSL task of specific test samples [18]. This overfitting hinder the model's performance on the main task. When test-time training is applied to rumor detection, the problem becomes even more precarious because rumor samples not only exhibit significant variations in their post attributes but also in the structure of the rumor propagation graphs. To alleviate this problem, we propose incorporating a constraint into the objective function during the test-time training phase. The core idea is to impose constraints on the embedding space generated by the shared feature extractor during test-time training, ensuring that the feature distribution of test examples remains close to that of the training domain.

The shared feature extractor consists of a 2-layers GCN. Let $H_1^2, H_2^2, \ldots, H_M^2$ represent the node embeddings that the shared graph feature extractor outputs for graphs $G_1, G_2, \ldots, G_M$ in the source domain data during training. After completing the training process, we obtain graph embeddings using a read-out function $h_i^2 = READOUT(H_i^2)$ and then compute two statistics based on these graph embeddings: the empirical mean $\mu_s = \frac{1}{M}\sum_i^M \mathbf{h}_i^2$

**Table 2: The experimental results on the Target(Source) domain. The best performances are highlighted in bold.**

| Target(Source) | Weibo-COVID19(Twitter) | | | | Twitter-COVID19(Weibo) | | | |
|---|---|---|---|---|---|---|---|---|
| Models | ACC. | Mac-F1 | RF1 | NF1 | ACC. | Mac-F1 | RF1 | NF1 |
| CNN | 0.445 | 0.402 | 0.476 | 0.328 | 0.498 | 0.389 | 0.528 | 0.249 |
| RNN | 0.463 | 0.414 | 0.498 | 0.329 | 0.510 | 0.388 | 0.533 | 0.243 |
| RvNN | 0.514 | 0.482 | 0.538 | 0.426 | 0.540 | 0.391 | 0.534 | 0.247 |
| PLAN | 0.532 | 0.496 | 0.578 | 0.414 | 0.573 | 0.423 | 0.549 | 0.298 |
| BiGCN | 0.569 | 0.508 | 0.586 | 0.429 | 0.616 | 0.415 | 0.577 | 0.252 |
| ACLR-BiGCN | 0.873 | 0.861 | 0.896 | 0.827 | 0.765 | 0.686 | **0.766** | 0.605 |
| T³RD(Ours) | **0.896** | **0.880** | **0.917** | **0.843** | **0.781** | **0.696** | 0.706 | **0.685** |

and the covariance matrix $\Sigma_s = \frac{1}{M-1}\left(H^{2^T}H^2 - \left(I^T H^2\right)^T\left(I^T H^2\right)\right)$, where $H^2 = \left\{h_1^{2^T}, \ldots, h_M^{2^T}\right\}$. During the test-time training process, when provided with an input graph $G_i$ from the test dataset $D^t$, we also compute embedding statistics for $G_i$ and its augmented views, denoting them as $\mu_t$ and $\Sigma_t$. The feature alignment aims to enforce the embedding statistics of the test graph sample to closely align with those of the training graph samples. This can be formally defined as:

$$L_a = \|\mu_s - \mu_t\|_2^2 + \|\Sigma_s - \Sigma_t\|_F^2. \quad (15)$$

where $\|\cdot\|_2$ is the Euclidean norm and $\|\cdot\|_F$ is the Frobenius norm.

## 5 EXPERIMENT

In this section, we conduct a series of experiments to assess the effectiveness of our proposed rumor detection model, T³RD. First, we evaluate its empirical performance by comparing it with several baseline models. Then, we analyze the impact of each variant of our model. After that, we investigate the capabilities of early rumor detection for both T³RD and the compared model. Finally, we provide extensive ablation studies to assess the effects of modules within the proposed model.

### 5.1 Datasets

We assess our proposed model, T³RD, using two widely used sets of real-world cross-domain rumor datasets. The first set encompasses the English Twitter dataset [21] and the Chinese Weibo-COVID19 dataset [14]. The second set comprises the Chinese Weibo dataset [20] and the English Twitter-COVID19 dataset [14]. These cross-domain datasets are annotated with two binary labels: Non-rumor (N) and Rumor (R). Detailed statistics for both sets of cross-domain datasets are provided in Tab. 1.

### 5.2 Baseline and Evaluation Metrics

We compare our model with the following baselines:
**CNN** [36]: A CNN-based approach for misinformation identification that frames the relevant posts as fixed-length sequences.
**RNN** [20]: An RNN-based rumor detection model with GRU used for feature learning from relevant posts over time.
**RvNN** [23]: A rumor detection model based on tree-structured recursive neural networks, which learns rumor representations guided by the propagation structure.

**PLAN** [12]: A transformer-based model designed for rumor detection to capture long-distance interactions between any pair of involved tweets.
**BiGCN** [2]: A GCN-based model that leverages directed conversation trees to learn higher-level representations (see Section 4.2).
**ACLR-BiGCN** [14]: An adversarial contrastive learning framework built on top of BiGCN. The model aims to bridge low-resource gaps for rumor detection by adapting features learned from well-resourced data to those of low-resource breaking events.
**RPL** [16]: A zero-shot rumor detection model based on prompt learning employs a hierarchical prompt encoding mechanism to acquire language-agnostic context representations for both prompts and rumor data.

We employ commonly-used metrics to evaluate the effectiveness of our proposed method. The accuracy (ACC) measures the probability of correctly predicting the samples. The F1-score provides distinct scores for positive (RF1), negative (NF1), and macro-average (Mac-F1). These metrics span a range from 0 to 1, with higher values signifying superior performance. We conduct 5-fold cross-validation on the target datasets.

### 5.3 Overall Performance

In Tab. 2, we evaluate our proposed method by comparing it with other methods tested on the Weibo-COVID19 and Twitter-COVID19 test sets. We note that the baselines in the first group exhibit poor performance, likely due to their sole reliance on sequential information while neglecting the propagation structure. Other baselines leverage the structural properties of information propagation on social media, affirming the significance of propagation structure representations within our framework. In the second group of structure-based baselines, PLAN and BiGCN outperform RvNN, attributed to the feature vector architecture and tree structures. The third group focuses on cross-domain rumor detection, with ACLR aligning the source and target domains through supervised contrast [14]. In contrast, our model T³RD employs test-time training to further extract additional information from the test data. In contrast, our proposed T³RD achieves state-of-the-art performance on the Weibo-COVID19 (Twitter-COVID19) dataset and makes a 2.3%(1.6%) improvement in accuracy scores, respectively. These results further emphasize the efficacy of test-time training in improving cross-domain rumor detection by extracting information from test data. In addition, we carry out experiments on zero-shot target

**Table 3: The target dataset is only used for testing rumor detection results. The best performances are highlighted in bold.**

| Target(Source) | Weibo-COVID19(Twitter) | | | |
|---|---|---|---|---|
| Models | ACC. | Mac-F1 | RF1 | NF1 |
| BiGCN (No-test) | 0.615 | 0.524 | 0.729 | 0.319 |
| ACLR_BiGCN (No-test) | 0.721 | 0.685 | 0.788 | 0.582 |
| RPL | 0.745 | 0.719 | 0.804 | 0.634 |
| T³RD (No-test) | **0.797** | **0.788** | **0.832** | **0.743** |

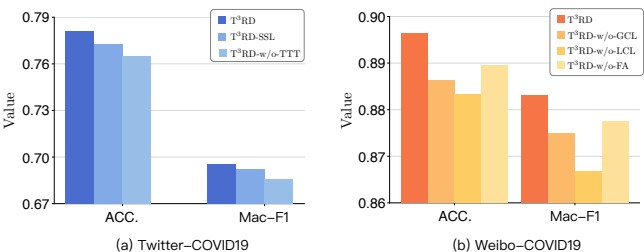

**Figure 3: The ablation study of components in T³RD.**

domain rumor detection, where the model is trained on source domain data and tested on target domain data. The experimental results are shown in Tab. 3. In comparison with the state-of-the-art, our proposed T³RD method achieves a 6.9% improvement in macro F1 score on the Weibo-COVID19 dataset. It proves the generalization capabilities of our methods.

## 5.4 Ablation Study

**Effectiveness of test-time training.** In Fig. 3 (a), we investigate the effectiveness of test-time training on rumor detection, in where T³RD represents our primary model, T³RD-SSL signifies the model with self-supervised learning incorporated during the training phase, and T³RD-w/o-TTT denotes the rumor detection model without test-time training. We adopt Weibo as the source domain dataset and Twitter-COVID19 as the target domain dataset. As depicted in Fig. 3 (a), performance exhibits a gradual decline, indicating the efficacy of self-supervised learning in rumor detection, while underscoring the importance of test-time training in information extraction from the test set.

**Effectiveness of components in T³RD.** In this subsection, we utilize Twitter as the source domain dataset and Weibo-COVID19 as the target domain dataset to evaluate the effectiveness of the components of our T³RD. As shown in Fig. 3 (b), we assess the impact of global contrastive learning, local contrastive learning, and feature alignment by eliminating each of them from T³RD. These three variants of T³RD are denoted as T³RD-w/o-GCL, T³RD-w/o-LCL, and T³RD-w/o-FA. The results indicate that the removal of any of these components results in a decline in model performance, underscoring the indispensability of the three components for effective rumor detection.

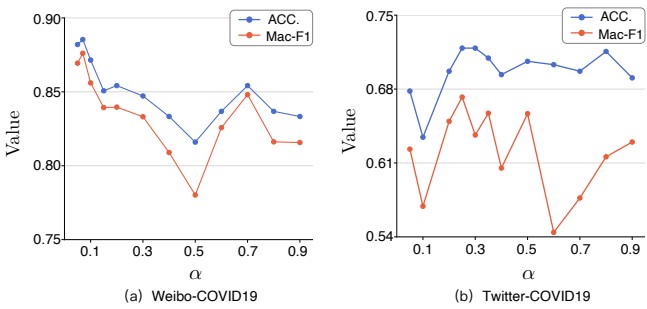

**Figure 4: Parameter analysis for the $\alpha$ on Weibo-COVID19 and Twitter-COVID19.**

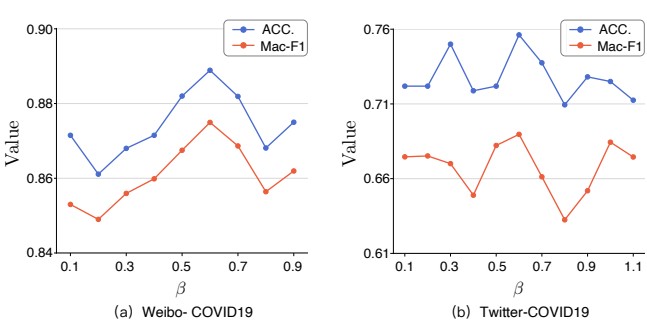

**Figure 5: Parameter analysis for the $\beta$ on Weibo-COVID19 and Twitter-COVID19.**

## 5.5 Hyper-parameter Analysis.

**Hyper-parameter $\alpha$.** To investigate the influence of the hyperparameter $\alpha$ in Eq. 2 on performance (Accuracy and Macro F1 score), we conduct a qualitative analysis within the T³RD architecture. For the target data Weibo-COVID19, we use Twitter as the source data (in Fig. 4 (a)). In terms of Twitter-COVID19, we use Weibo as the source data (in Fig. 4 (b)). The horizontal axis represents the values of $\alpha$. We observe that when Weibo-COVID19 is the target dataset, the optimal result is achieved at $\alpha = 0.07$, and for Twitter-COVID19, the best result is obtained at $\alpha = 0.25$. From the overall trend, as the $\alpha$ value increases, the performance exhibits a certain degree of fluctuation. We believe this occurs because the model while optimizing the representation distribution, compromises the mapping relationship with labels.

**Hyper-parameter $\beta$.** To study the effects on performance (Accuracy and Macro F1 score) of the hyper-parameter $\beta$ in our Eq. 14, we conduct qualitative analysis under T³RD architecture(in Fig. 5). The horizontal axis denotes the value of $\beta$. According to the results, we observe that the optimal outcome is achieved at $\beta = 0.6$ when Weibo-COVID19 is the target dataset, while $\beta = 0.6$ yields the best result when Twitter-COVID19 is the target dataset.

**Hyper-parameter $\gamma$.** To study the Effectiveness of hyper-parameter $\gamma$ in our Eq. 3, we conduct qualitative analysis under T³RD architecture in Fig. 6. The horizontal axis denotes the value of $\gamma$. According to the results, we observe that the optimal outcome is achieved at

$\gamma = 0.7$ when Weibo-COVID19 is the target dataset, while $\gamma = 0.5$ yields the best result when Twitter-COVID19 is the target dataset.

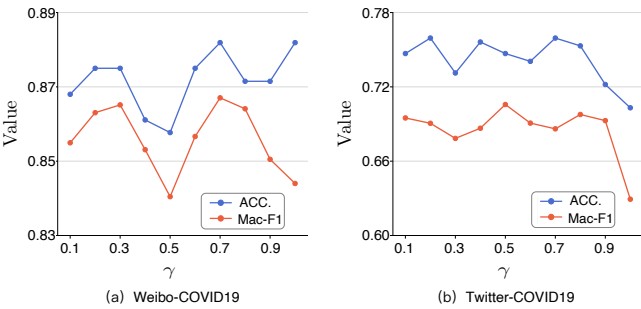

(a) Weibo-COVID19      (b) Twitter-COVID19

**Figure 6: Parameter analysis for the $\gamma$ on Weibo-COVID19 and Twitter-COVID19.**

## 5.6 Early Detection

Early detection aims to identify rumors in the early stages of their propagation, which is crucial for minimizing their societal impact. Therefore, it serves as another important metric for evaluating the quality of rumor detection models. To construct an early detection task, we establish a series of "delays" detection checkpoints. We determine these checkpoints based on either the number of reply posts or the time that has elapsed since the initial post. Only content posted no later than the checkpoint time is used for model evaluation. Performance is assessed using macro F1 scores obtained at each checkpoint.

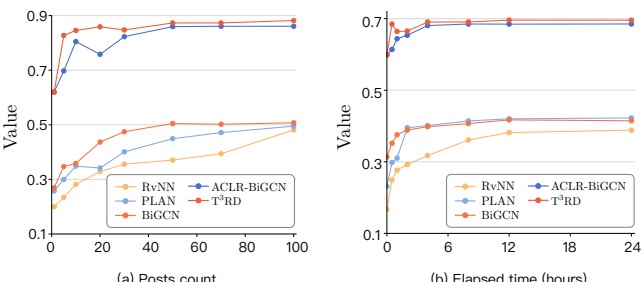

(a) Posts count      (b) Elapsed time (hours)

**Figure 7: Early detection performance is assessed at various checkpoints based on the count of posts (or elapsed time) on both the Weibo-COVID19 (a) and Twitter-COVID19 (b) datasets.**

In Fig. 7, we make a comparison of the performance between our method with RvNN, PLAN, BiGCN, and ACLR at various checkpoints. The proposed T$^3$RD outperforms other approaches throughout the entire lifecycle and achieves a relatively high Macro F1 score at an early stage. Our method requires about 20 posts on Weibo-COVID19 and 4 hours on Twitter-COVID19 to achieve stable performance, while the state-of-the-art method ACLR requires 50 posts to achieve a similar level of performance. This demonstrates the outstanding early detection capability of our method.

**Table 4: Experimental results of Cross-domain Rumor Detection. The best performances are highlighted in bold.**

| Target | Weibo-COVID19 | | Twitter-COVID19 | |
|---|---|---|---|---|
| Models | ACC. | Mac-F1 | ACC. | Mac-F1 |
| ACLR(cross-D) | 0.884 | 0.855 | 0.737 | 0.623 |
| T$^3$RD(cross-D) | **0.900** | **0.889** | 0.771 | 0.689 |
| T$^3$RD(cross-D&L) | 0.896 | 0.880 | **0.781** | **0.696** |

Additionally, early-stage performance tends to exhibit more or less fluctuation. This is due to the increase in semantic and structural information as statements propagate, resulting in a corresponding increase in noise.

## 5.7 Cross-domain Rumor Detection

In this subsection, we exclusively conduct experiments within the cross-domain setting. When considering the Weibo-COVID19 as the target data, we employ the WEIBO dataset with comprehensive annotation as the source data. Similarly, for the Twitter-COVID19, we designate the TWITTER dataset as the source data. The results are shown in Tab. 4. With Weibo as the source data, our model improves the 0.4% Accuracy and 0.9% Macro F1 on the Weibo-COVID19 dataset, indicating our superior ability in cross-domain rumor detection on Weibo. However, the overall performance on Twitter-COVID19 is relatively worse with Twitter as the source dataset. The reason may be that the number of events in the Twitter dataset is smaller than in Weibo, in which our model could achieve about 77.1% Accuracy and 68.9% Macro F1 score among the variants of response ranking. The proposed T$^3$RD could alleviate the low-resource issue of rumor detection as well as diminish the need for extensive reliance on datasets annotated with specific domain and language knowledge, which enables to leverage the knowledge from Weibo instead of just Twitter to detect rumors in Twitter-COVID19 for better performance.

## 6 CONCLUSION AND FUTURE WORK

In this work, we design a test-time training framework for cross-domain rumor detection. This framework introduces an additional test-time training phase between the training and testing phases, aiming to bridge the performance gap between the training set and the test set. In the test-time training phase, we employ self-supervised learning to extract information from the test set. It comprises global and local contrastive learning. Global contrastive learning captures invariant graph representations that are integrated into node representations, while local contrastive learning acquires invariant node representations. Comprehensive experimental results on two widely used datasets demonstrate the superiority of the proposed method compared with other state-of-the-art rumor detection methods. In future work, we will make the exploration of design a more effective auxiliary task for test-time training on rumor detection. We also would like to explore the generalization of rumor detection in other domains and minority languages.

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
