# OpenReview forum: "T$^3$RD: Test-Time Training for Rumor Detection on Social Media"
_ACM.org/TheWebConf/2024/Conference — TheWebConf24 Oral_

### Official Review · Reviewer_w9xs · 2023-10-26

**Novelty:** 6
**Technical Quality:** 5

**Review:**

This work aims to bridge the low-resource gap for rumor detection on social media, from a fresh perspective of test-time training, for which a  novel model is proposed to boost classification performance and mitigate the potential risk of feature distortion, with various self-supervised learning tasks elaborately designed. Extensive experiments on real-world low-resource rumor data confirm the effectiveness of the proposed approach.

- Pros:
1) The low-resource rumor detection task is a promising but urgent problem in the era of social media. The scope of this work is meaningful and closely relevant to the Web.
2) Approaching this low-resource rumor detection problem from the perspective of test-time training is both novel and carefully designed, rather than simply stacking technologies.
3) The experiment results and discussion are sufficient and comprehensive enough to support the stated claims in this manuscript for the few-shot and zero-shot transfer.
4) The paper is well-written and well-organized.

- Cons:
1) I understand the test-time training framework is designed for the graph-based backbone model, if tested on more GNN-based rumor detection models, the wide scalability of the proposed framework would be better illustrated.
2) There is a minor typo in Sec.4.3.3: Through minimizing the SSL loss defined in Eq. ?.

**Questions:**

Why not further conduct the zero-shot rumor detection with the Weibo dataset as the source and the Twitter-COVID19 dataset as the target?

**Reviewer Confidence:**

4: The reviewer is certain that the evaluation is correct and very familiar with the relevant literature

**Scope:**

4: The work is relevant to the Web and to the track, and is of broad interest to the community

---

### Official Review · Reviewer_Fop6 · 2023-11-16

**Novelty:** 6
**Technical Quality:** 6

**Review:**

Summary:

This work introduces T3RD, which aims to enhance the performance of rumor detection models on low-resource datasets. The approach employs SSL as an auxiliary task in the test-time training phase, which helps to calibrate the trained model for test samples. The paper also introduces a feature alignment technique to achieve a balanced synergy between the knowledge derived from the training set and the test samples.

Strengths:

S1. This work introduces a novel approach toward rumor detection on low-resource datasets.

S2. The comprehensive experimental results on two, previously used by other works, datasets demonstrate better performance compared with other state-of-the-art rumor detection methods.

S3. To me, this seems like a complete paper: well-organized, clearly presents the proposed methods and results, and detailed explanations of the methodology and results.

Weaknesses:

Here, I list some thoughts I had while reading the paper. Please refer to the comments below for a more detailed explanation.

W1. The problem is very specific, and it is not clear whether this model would work well in detecting other kinds of rumors online. More specifically, there is no discussion from the authors on whether this approach would work on a universal problem/dataset.

W2. I would expect more comparisons with other state-of-the-art rumor detection models that the authors may have neglected.

W3. The authors do not address reproducibility in their manuscript.

Comments:

I thank the authors for submitting their manuscript to WWW. I enjoyed reading their work. Please review the following comments as constructive criticism of your work toward making the manuscript stronger.

C1: Regarding point W1, my main issue is that the authors focus on a very specific problem, using very 'narrow' datasets. I would expect some discussion regarding the applications of the method proposed in this paper and how well it would perform on other rumor detection. Follow up in Comment 2.

C2: I understand that this manuscript proposes a 'proof of concept' regarding a more accurate method of detecting rumors, but considering that Twitter is no longer open for researchers for free, how do the authors expect this model to work on other forms of data, either longer or shorter, similarly to Reddit-like data that is still open to researchers?

C3: Regarding W2, the authors did well comparing their work with the state of the art, but I was surprised there was no mention of BERT and RoBERTa, or adversarial training for rumor detection online. Please see [1][2][3], and [4] below for some examples. In case these models are not relevant to this work, could the authors elaborate on why?

C4: Lastly, regarding W3, I wonder whether this work is easy to reproduce and what efforts the authors make to ensure their work is transparent. For example, will this model be released after publication? Is there documentation regarding the specific fine-tuning the authors did that is not included in the paper? For example, the paper does not provide a detailed analysis of the impact of different hyperparameters on the performance of the proposed method. Would a documentation/appendix include this information in the camera-ready version in case this work moves to publication?

C5: The authors scrutinize other models as not being multilingual. This work is trained on labeled data for English and Chinese. That means that the model is able to detect rumors in those 2 languages, but they then never refer to these issues again. What about other languages?

Other minor comments:
For readability purposes, I suggest the authors include the names of the authors they refer to when they cite their work. For example, "John et al.[2]" did so and so, instead of "[2] did so and so"

[1] Yuechen, Li, Qian Lingfei, and Ma Jing. "Early Detection of Rumors Based on BERT Model." AI and Analytics for Public Health: Proceedings of the 2020 INFORMS International Conference on Service Science. Springer International Publishing, 2022.

[2] Fan, Guowei, Tao Wu, and Yamei Lei. "Research on Rumor Detection Based on RoBERTa-BiGRU Model." International Conference on Artificial Intelligence and Security. Cham: Springer International Publishing, 2022.

[3] Ni, Shiwen, Jiawen Li, and Hung-Yu Kao. "HAT4RD: Hierarchical Adversarial Training for Rumor Detection in Social Media." Sensors 22.17 (2022): 6652.

[4] Zhang, Taozheng, and Shuaidong Hu. "Unsupervised Cross-Domain Rumor Detection from Multiple Sources Based on RoBERTa and Multi-CNN." Proceedings of the 2023 7th International Conference on Deep Learning Technologies. 2023.

**Questions:**

For the rebuttal, I'd appreciate the authors' input regarding Comments C1, C2, C3, C4, and C5, above.

**Ethics Review Description:**

I do not detect Ethical issues with this work

**Reviewer Confidence:**

2: The reviewer is willing to defend the evaluation, but it is likely that the reviewer did not understand parts of the paper

**Scope:**

4: The work is relevant to the Web and to the track, and is of broad interest to the community

---

### Official Review · Reviewer_yr56 · 2023-11-17

**Novelty:** 5
**Technical Quality:** 4

**Review:**

Strengths:

    Novelty and Originality: The approach adopted in this study is commendable for its novelty and originality. The unique methodology and the intriguing results contribute significantly to the field.
    Documentation: The authors have done an excellent job in documenting their approach thoroughly. This level of detail enhances the reproducibility and understanding of the study.

Weaknesses:

    Dataset Selection Justification: The choice of datasets, particularly their relevance to COVID-19, is inadequately explained. A more detailed justification is needed, especially in terms of how this choice impacts the generalizability of the results to other contexts or datasets.
    Definition and Operationalization of Rumors: The paper lacks clarity on what constitutes a 'rumor.' It's crucial to explicitly define rumors and differentiate them from general misinformation. How rumors are operationalized in this study also needs clearer articulation.
    Cross-Domain Detection and Societal Contexts (Section 5.7): There is a noticeable gap in the discussion regarding how the model's performance in cross-domain detection might be influenced by variations in textual data arising from different societal contexts. Addressing this could significantly strengthen the paper, providing insights into the model's adaptability and reliability across diverse datasets.

**Questions:**

A more comprehensive explanation of why specific datasets were chosen and how they align with the study's objectives would be beneficial. Discussing their limitations and how they impact generalizability could improve the paper.

 Providing clear definitions and operational criteria for key terms like 'rumors' and 'misinformation' would greatly aid in understanding the study's focus as well as assessing validity and generalizability of the results.

A deeper analysis or discussion on how societal contexts influence model performance in cross-domain detection would be a valuable addition. This could involve exploring linguistic or cultural factors that might affect the detection accuracy.

**Reviewer Confidence:**

2: The reviewer is willing to defend the evaluation, but it is likely that the reviewer did not understand parts of the paper

**Scope:**

3: The work is somewhat relevant to the Web and to the track, and is of narrow interest to a sub-community

---

### Official Review · Reviewer_FtfK · 2023-11-22

**Novelty:** 6
**Technical Quality:** 5

**Review:**

Test time training regards a test dataset as an unlabeled dataset, uses the dataset to adjust the encoder via a pretext self-supervised task with global contrastive loss and local contrastive loss. This process is similar with indusive classification where test  samples without labels can be seen in the training process. This strategy did not studied in the field of rumor detection. The experimental tests show that this strategy is very helpful for improving the performance of rumor detection in low-resource cross-domain scenarios. Thus, this study deserves to be carried out.

The manuscript still can be improved in the following.
1. In equation 3, it is not clear what is L_a. We can see it is appeared in equation 15, but it had better be defined when it is used.
2.  In equation 11, it is not clear what is s.
3. The mathematical symbols in the manuscript are not well organized. Latex is suggested to be used to edit the manuscript.
4. Zero-shot target domain experiments had better be also performed on Twitter-COVID19(Weibo) besides Weibo-Covid19 (Twitter) showed in Table 3.

**Questions:**

The problem 4 had better be answered.

**Reviewer Confidence:**

4: The reviewer is certain that the evaluation is correct and very familiar with the relevant literature

**Scope:**

4: The work is relevant to the Web and to the track, and is of broad interest to the community

---

### Official Review · Reviewer_vFbc · 2023-11-22

**Novelty:** 5
**Technical Quality:** 4

**Review:**

This paper studies on how to improve the rumor detection for low-resource emergent events by using test-time training. Specifically, the authors designed a test-time training (or we can also call it as domain-adaptation method) framework that design a label-free auxiliary task that can be trained on both training domain and testing domain to improve the testing domain accuracy. This framework contains local and global contrastive learning to adapt the spreading process of a rumor. The strengths and weakness include:

Strength:

The idea of focusing on both local and global information is interesting.

Weakness:

My main concern is that the experiment of comparison is problematic. The key contribution of this paper, as claimed by the authors, is the proposed test-time training framework. Therefore, the comparison should focus on the performance boost brought by the training framework and should be invariant to the backbone. Thus, a more reasonable comparison strategy should be using the same backbone (e.g., the backbone of proposed model or the BiGCN) and compare their performance with different test-time training strategies (e.g., ACLR). For example, in the paper of ACLR, which is the main baseline in this submission, the authors of ACLR compare ACLR-RvNN, ACLR-PLAN and ACLR-BiGCN with DANN plusing all these backbones. However, in this paper, the authors compare their model and baselines in a way that we can not identify whether the boost is from the training strategy or the backbone. This problem is critical, because if we check Table 2 and 3 in details, we will see that actually the boost that ACLR brought to BiGCN is larger than the boost that their test-time-training strategy brought to their GCN backbone.

**Questions:**

Please response to the weakness with experimental results.

**Reviewer Confidence:**

3: The reviewer is confident but not certain that the evaluation is correct

**Scope:**

3: The work is somewhat relevant to the Web and to the track, and is of narrow interest to a sub-community

---

### Decision · Program_Chairs · 2024-01-22

**Decision:**

Accept (Oral)

**Comment:**

The purpose of the paper is tackling the problem of rumor detection on new (small domain) topics. The main tools proposed for this self-supervised learning and a local/global contrastive approach.
 Overall, the reviewers appreciated the work. There were a number of comments about reproducibility, baselines and lack of clarity, but all of them were satisfactorily answered in the rebuttal phase.
 I believe that the paper can be accepted, provided that the final version includes the most important answers provided in the rebuttal, and that the code be made available as promised.